# Synthesis and Antibacterial Evaluation of Novel 1,3,4-Oxadiazole Derivatives Containing Sulfonate/Carboxylate Moiety

**DOI:** 10.3390/molecules25071488

**Published:** 2020-03-25

**Authors:** Lei Wang, Xia Zhou, Hui Lu, Xianfu Mu, Linhong Jin

**Affiliations:** State Key Laboratory Breeding Base of Green Pesticide and Agricultural Bioengineering, Key Laboratory of Green Pesticide and Agricultural Bioengineering, Ministry of Education, Guizhou University, Huaxi District, Guiyang 550025, China; wanglei880328@163.com (L.W.); luhui2624904231@163.com (H.L.); Mu_xianfu@163.com (X.M.)

**Keywords:** 1,3,4-oxadiazole derivatives, antibacterial activity, *xanthomonas oryzae* pv. *oryzae*, *xanthomonas axonopodis* pv. *Citri*, scanning electron microscopy

## Abstract

In order to discover new lead compounds with high antibacterial activity, a series of new derivatives were designed and synthesized by introducing a sulfonate or carboxylate moiety into the 1,3,4-oxadiazole structure. Antibacterial activity against two phytopathogens, *Xanthomonas oryzae* pv. *oryzae (Xoo)* and *Xanthomonas axonopodis* pv. *citri*
*(Xac)*, was assayed in vitro. The preliminary results indicated that ten compounds including **4a-1-4a-4** and **4a-11-4a-16** had good antibacterial activity against *Xoo*, with EC_50_ values ranging from 50.1-112.5 µM, which was better than those of Bismerthiazol (253.5 µM) and Thiodiazole copper (467.4 µM). Meanwhile, **4a-1**, **4a-2**, **4a-3** and **4a-4** demonstrated good inhibitory effect against *Xanthomonas axonopodis pv. citri* with EC_50_ values around 95.8-155.2 µM which were better than those of bismerthiazol (274.3 µM) and thiodiazole copper (406.3 µM). In addition, in vivo protection activity of compound **4a-2** and **4a-3** against rice bacterial leaf blight was 68.6% and 62.3%, respectively, which were better than bismerthiazol (49.6%) and thiodiazole copper (42.2%). Curative activity of compound **4a-2** and **4a-3** against rice bacterial leaf blight was 62.3% and 56.0%, which were better than bismerthiazol (42.9%) and thiodiazole copper (36.1%). Through scanning electron microscopy (SEM) analysis, it was observed that compound **4a-2** caused the cell membrane of *Xanthomonas oryzae* pv. *oryzae* ruptured or deformed. The present results indicated novel derivatives of 5-phenyl sulfonate methyl 1,3,4-oxadiazole might be potential antibacterial agents.

## 1. Introduction

Bacterial diseases of rice plants will lead to the reduction of rice yield and hence serious decreases in crop quality and insufficient food supply [1,2,3]. Bacterial leaf blight of rice infected by *Xanthomonas oryzae* pv. *oryzae (Xoo)* will reduce rice yield by affecting rice growth [4,5]. Citrus canker, the devastating citrus disease caused by *Xanthomonas axonopodis* pv. *citri (Xac)*, can severely affect citrus production [6,7]. Bismerthiazol (BT) and thiodiazole copper (TC) are traditional systemic fungicides, which are commonly used to treat rice bacterial leaf blight and citrus canker [8,9]. However, the long-term frequent application of them has led to bactericide-resistant, therefore the phenomenon that rice bacterial leaf blight and citrus canker cannot be effectively controlled has emerged [10]. So, it is urgent to develop efficient new chemical pesticides to deal with this problem.

We previously found that 1,3,4-oxadiazole derivatives have a variety of biological effects, including antibacterial [11,12,13], antifungal [14,15], antiviral [16], nematocidal [17] and insecticidal [18,19] activity. 1,3,4-oxadiazole has the ideal heterocyclic structure to be developed into efficient pesticides. Meantime, sulfonate or carboxylate derivatives have broad-spectrum biological activity in agriculture, such as insecticidal [20], antibacterial [21], antiviral [22] and antifungal [23] activity. For example, pyraoxystrobin [24], chlorfenson [25] and nimrod [26] (Figure 1) containing sulfonate or carboxylate respectively have been widely used in agriculture [27,28,29].

In addition, we reported the splicing of oxymethyl and 1,3,4-oxadiazole sulfone derivatives could provide excellent antibacterial activity [10,30]. Based on those prior works, we hypothesized that sulfonate/carboxylate moiety functionalized 1,3,4-oxadiazole derivatives might also show promising antibacterial activity. Hence in the present work, a series of novel compounds were synthesized by introducing sulfonate or carboxylate moiety to 1,3,4-oxadiazole to discover new structures with potential antibacterial activity. The design and synthesis of the targets are depicted in Figure 1 and Scheme 1 respectively.

## 2. Results and Discussion

### 2.1. Chemistry

As described in Scheme 1, starting from ethyl glycolate, the key intermediate (5-mercapto-1,3,4-oxadiazol-2-yl) methanol **2** was synthesized in two steps involving acylation and cyclization. Subsequently, intermediate **2** was converted into its corresponding thioether derivative **3** by thioetherification with R_1_I. Finally, the target compounds **4a/5a** was obtained by esterification with R_2_SOOCl/R_3_COCl. The structures of all target compounds were confirmed by nuclear magnetic resonance spectra including ^1^H NMR, ^13^C NMR and electrospray ionization high-resolution mass spectrometry (ESI-HRMS). Fluorine nuclear magnetic resonance (^19^F NMR) was involved for some fluoride structures.

### 2.2. In Vitro Antibacterial Activity

The antibacterial activity of all the compounds was evaluated in vitro against *Xanthomonas oryzae* pv. *oryzae (Xoo)* and *Xanthomonas axonopodis* pv. *(Xac)* via the turbidimeter test [10]. Bismerthiazol and thiodiazole copper served as positive controls to compare the bactericidal potency of the tested compounds.

As shown in Table 1, most of the compounds **4a** exhibited higher antibacterial activity than either bismerthiazol or thiodiazole copper against the tested plant bacteria. Among them, inhibitory rates for *Xoo* of compounds **4a-1–4a-5** and **4a-11-4a-16** at 100 µg/mL as well as **4a-1**, **4a-2**, **4a-14** at 50 µg/mL were all above 90%. Inhibitory rates for *Xac* of compounds **4a-1**–**4a-3**, 93%–97% at 200 µg/mL and 69%–82% at 100 µg/mL were also superior to those of positive controls. At the same time, the present tests were parallelly conducted on compounds **5a**. Actually, no similar tendency was observed on **5a** against tested bacteria. It was confirmed that compounds **4a** bearing sulfonate moiety were more potent in combating *Xoo* and *Xac* and presented remarkable higher activity as compared to compounds **5a** and positive controls.

Further, compounds acting better than positive controls bismerthiazol or thiodiazole copper (Table 1) were performed for their EC_50_ values (Table 2). Compounds **4a-1-4a-4** and **4a-11-4a-16** revealed outstanding activity against *Xoo* with EC_50_ values around 50.1–112.5 µM, which was lower than bismerthiazol (253.5 µM) and thiodiazole copper (467.4 µM). Meanwhile, EC_50_ (95.8–155.2 µM) values of **4a-1-4a-4** against *Xac* were also lower than 274.3 µM displayed by bismerthiazol and 406.3 µM by thiodiazole copper.

In particular, **4a-2** bearing 4-F substituted benzenesulfonate, performed the best on *Xoo* and *Xac* with EC_50_ values of 50.1 and 95.8 µM respectively, which were quite better than two commercial positive controls. So, compound **4a-2** appeared to be promising antibacterial agents against plant bacterial diseases.

### 2.3. In Vivo Antibacterial Activity

With outstanding bactericidal activity of compounds **4a-1, 4a-2, 4a-3** in vitro, they were further explored for their antibacterial potency in vivo against rice bacterial leaf blight via the leaf-cutting method [10]. Bismerthiazol and thiodiazole copper served as positive controls for this investigation. All inoculated plants in 14 days exhibited blight symptoms with 100% morbidity.

At the concentration of 200 µg/mL, as shown in Figure 2 and Table 3, the control efficiency of the protection activity of compounds **4a-2** and **4a-3** were 68.6% and 62.3%, which were superior to Bismerthiazol (49.6%) and Thiodiazole copper (42.2%). As shown in Figure 3 and Table 4, the control efficiency of the curative activity of compound **4a-1**, **4a-2** and **4a-3** were 44.6%, 62.3% and 56.0%, which were superior to bismerthiazol (42.9%) and thiodiazole copper (36.1%).

### 2.4. Scanning Electron Microscopy Studies

Scanning electron microscopy offers the ability to observe the bacterial cell surface [30]. Based on the analysis of antibacterial against *Xoo* results in vitro and in vivo, the antibacterial mechanism of compound **4a-2** was studied by SEM. As shown in the Figure 4, when the compound **4a-2** was at a concentration of 25 µg/mL, the bacterial cell was deformed, and part of the bacterial cell wall was slightly ruptured. When the compound **4a-2** concentration was increased to 50 µg/mL, most cell membrane were wrinkled and ruptured. Then observing the control group (A), these bacterial cells were round and smooth, without any breakage. Scanning electron microscopy images had further demonstrated that the compounds **4a-2** have antibacterial activity against *Xoo*.

### 2.5. Structure-Activity Relationship (SAR) Analyses

Based on the activity values shown in Table 1 and Table 2, a preliminary conclusion could be drawn about the structure-activity relationship. First, according to the antibacterial research of those **4a** and **5a** derivatives, it had shown that compound **5a** derivatives containing the sulfonate structure was significantly higher efficient than the corresponding **5a** containing the carboxylate structure. Obviously, the existence of the sulfonate structure was very important to improve inhibitory effect.

Further antibacterial evaluation on *Xoo* and *Xac* showed that 4-substituted halogenated phenyl sulfonate derivatives expressed significant antibacterial activity. Three (**4a-2**, **4a-3**, **4a-4**) of them worked well on both *Xoo* and *Xac*, which appeared an obvious decreasing potency (50.1, 87.2, 99.4 µM) with increasing halogen size in **4a-2**(R_2_ = F), **4a-3**(R_2_ = Cl), **4a-4**(R_2_ = Br) respectively. In this regard, it was consistent with previous reports [10]. The other six compounds **4a-11-4a-16** (R_1_ = C_2_H_5_) also showed extensive potency on the *Xoo*. However, their EC_50_ are slightly decreased like 98, 95, 86.4 µM and not necessarily following the tendency **4a-11**(F) >**4a-12**(Cl) >**4a-13**(Br). It can be refereed that R_1_ in thioether side chain also make difference in the activity of the structure. So in particular, when R_1_ = CH_3_, R_2_ = F, compound **4a-2** would be the most promising compound both in vitro and in vivo against the tested plant bacteria.

## 3. Experimental

### 3.1. Chemicals and Instruments

All reagent products from the Chinese Chemical Reagent Company were analytical or chemical pure. Thin-layer chromatography (TLC) of a GF254 silica gel pre-coated plate (Qingdao Haiyang Chemical Co., Ltd., Qingdao China) was used to evaluate the progress of the reaction and the purity of the compounds. Melting points were determined using an XT-4 digital melting-point apparatus (Beijing Tech. Instrument Co., Beijing, China) and reading was uncorrected. ^1^H NMR, ^13^C NMR and ^19^F NMR spectra were recorded on a 400 MHz spectrometer (Swiss Bruker, Swiss, Germany) with CDCl_3_ or (CD_3_)_2_CO-*d_6_* as the solvent. The antibacterial mechanism was studied by scanning electron microscopy (FEI, Hillsboro Oregon, America). Single crystal structure was collected by single crystal diffractometer (Gemini E, Oxford, United Kingdom). High-resolution mass spectral (HRMS) data were performed with Thermo Scientific Q Exactive (Thermo, Waltham, MA, USA).

### 3.2. General Synthetic Procedure for the Target Compounds

#### 3.2.1. Preparation of Intermediate 1

Ethyl glycolate (0.05 mol) was dissolved with 100 mL ethanol in a round bottom flask. Then, 80% of hydrazine hydrate (0.1 mol) was slowly added to the round bottom flask at room temperature. After a day of reaction, white product **1** will precipitate out in 85%–90% yields.

#### 3.2.2. Preparation of Intermediate 2

To a three-necked round bottom flask was added intermediate **1** (0.01 mol), KOH (0.012 mol) and 100 mL of ethanol in this order. Then, carbon disulfide (0.012mol) was slowly added under a stirred condition. The mixture was reacted at room temperature for 1–2 h and then heated to 78 °C for refluxing of six hours. The solution was removed under reduced pressure on a rotary evaporator, and product **2** was obtained in 65%–70% yields.

#### 3.2.3. Preparation of Intermediate 3

Tetrahydrofuran was used for dissolving intermediate **2** (0.01 mmol), then added KOH (0.012 mmol) and R_1_I (0.012 mmol). The reaction was judged complete by TLC, the solvent was removed under reduced pressure, and the residue was purified by silica gel column chromatography to obtain Intermediate **3** in yield of 70%–80%.

#### 3.2.4. Preparation of Target Compound 4a/5a

At room temperature, added intermediate **3** (0.001 mol), tetrahydrofuran (10 mL), and sodium hydride (0.001 mol) to the round-bottomed flask in order. After stirring for 30 min, R_2_SOOCl/R_2_COCl (0.001 mol) was slowly added, and the reaction was followed by TLC and filtered to get **4a/5a**.

*(5-(methylthio)-1,3,4-oxadiazol-2-yl)methyl benzenesulfonate* (**4a-1)**. White solid; m.p.: 54–55 °C; yield, 80.5%; ^1^H NMR (400 MHz, CDCl_3_) δ 7.93 (dd, *J* = 8.4, 1.2 Hz, 2H, Ar-H), 7.70 (t, *J* = 7.5 Hz, 1H, Ar-H), 7.58 (t, *J* = 7.8 Hz, 2H, Ar-H), 5.23 (s, 2H, -CH_2_-), 2.69 (s, 3H, -CH_3_). ^13^C NMR (100 MHz, CDCl_3_) δ 167.60, 160.69, 135.10, 134.50, 129.49, 128.13, 59.75, 14.51. HRMS calculated for C_10_H_11_O_4_N_2_S_2_ [M + H]^+^ 287.01547, found 287.01529.

*(5-(methylthio)-1,3,4-oxadiazol-2-yl)methyl 4-fluorobenzenesulfonate* (**4a-2)**. White solid; m.p.: 64–65 °C; yield, 86.5%; ^1^H NMR (400 MHz, Acetone) δ 8.04 (dd, *J* = 9.0, 5.0 Hz, 2H, Ar-H), 7.48 (t, *J* = 8.8 Hz, 2H, Ar-H), 5.44 (s, 2H, -CH_2_-), 2.71 (s, 3H, -CH_3_). ^13^C NMR (100 MHz, Acetone) δ 166.81, 166.10 (d, *J* = 254.9 Hz), 161.21, 131.72 (d, *J* = 3.5 Hz), 131.30 (d, *J* = 10.2 Hz), 116.91 (d, *J* = 23.3 Hz), 60.57, 13.67. ^19^F NMR (376 MHz, Acetone) δ -104.45. HRMS calculated for C_10_H_10_FO_4_N_2_S_2_ [M + H]^+^ 305.00605, found 305.00592.

*(5-(methylthio)-1,3,4-oxadiazol-2-yl)methyl 4-chlorobenzenesulfonate* (**4a-3**). White solid; m.p.: 85–86 °C; yield, 86.5%; ^1^H NMR (400 MHz, CDCl_3_) δ 7.78 (d, *J* = 8.8 Hz, 2H, Ar-H), 7.47 (d, *J* = 8.8 Hz, 2H, Ar-H), 5.18 (s, 2H, -CH_2_-), 2.63 (s, 3H, -CH_3_). ^13^C NMR (100 MHz, CDCl_3_) δ 167.73, 160.51, 141.34, 133.69, 129.80, 129.53, 59.92, 14.52. HRMS calculated for C_10_H_10_O_4_N_2_ClS_2_ [M + H]^+^ 320.97650, found 320.97629.

*(5-(methylthio)-1,3,4-oxadiazol-2-yl)methyl 4-bromobenzenesulfonate* (**4a-4**). White solid; m.p.: 79–80 °C; yield, 76.5%; ^1^H NMR (400 MHz, CDCl_3_) δ 7.69 (d, *J* = 8.7 Hz, 2H, Ar-H), 7.64 (d, *J* = 8.7 Hz, 2H, Ar-H), 5.18 (s, 2H, -CH_2_-), 2.63 (s, 3H, -CH_3_). ^13^C NMR (100 MHz, CDCl_3_) δ 167.76, 160.48, 134.18, 132.80, 129.93, 129.55, 59.97, 14.55. HRMS calculated for C_10_H_10_O_4_N_2_BrS_2_ [M + H]^+^ 364.92599, found 364.92548.

*(5-(methylthio)-1,3,4-oxadiazol-2-yl)methyl 4-methoxybenzenesulfonate* (**4a-5**). White solid; m.p.: 63–64 °C; yield, 66.5%; ^1^H NMR (400 MHz, CDCl_3_) δ 7.85 (d, *J* = 9.0 Hz, 2H, Ar-H), 7.02 (d, *J* = 8.9 Hz, 2H, Ar-H), 5.19 (s, 2H, -CH_2_-), 3.90 (s, 3H, -CH_3_), 2.69 (s, 3H, -CH_3_). ^13^C NMR (100 MHz, CDCl_3_) δ 167.50, 164.34, 160.87, 130.46, 126.20, 114.67, 59.51, 55.83, 14.50. HRMS calculated for C_11_H_13_O_5_N_2_S_2_ [M + H]^+^ 317.02604, found 317.02472.

*(5-(methylthio)-1,3,4-oxadiazol-2-yl)methyl 4-nitrobenzenesulfonate* (**4a-6**). White solid; m.p.: 86–87 °C; yield, 74.5%; ^1^H NMR (400 MHz, CDCl_3_) δ 8.42 (d, *J* = 9.0 Hz, 2H, Ar-H), 8.12 (d, *J* = 9.0 Hz, 2H, Ar-H), 5.36 (s, 2H, -CH_2_-), 2.70 (s, 3H, -CH_3_). ^13^C NMR (100 MHz, CDCl_3_) δ 167.88, 160.22, 151.07, 140.95, 129.52, 124.59, 60.42, 14.50. HRMS calculated for C_10_H_10_O_6_N_3_S_2_ [M + H]^+^ 332.00055, found 332.00040.

*(5-(methylthio)-1,3,4-oxadiazol-2-yl)methyl 4-(trifluoromethyl)benzenesulfonate* (**4a-7**). White solid; m.p.: 60–61 °C; yield, 84.5%; ^1^H NMR (400 MHz, CDCl_3_) δ 7.98 (d, *J* = 8.2 Hz, 2H, Ar-H), 7.77 (d, *J* = 8.3 Hz, 2H, Ar-H), 5.23 (s, 2H, -CH_2_-), 2.61 (s, 3H, -CH_3_). ^13^C NMR (100 MHz, CDCl_3_) δ 166.76, 159.29, 137.84, 135.19, 134.86, 127.66, 125.55 (q, *J* = 3.6 Hz), 123.24, 120.53, 59.16, 13.39. ^19^F NMR (376 MHz, CDCl_3_) δ -63.36. HRMS calculated for C_11_H_10_O_4_N_2_F_3_S_2_ [M + H]^+^ 355.00286, found 355.00241.

*(5-(methylthio)-1,3,4-oxadiazol-2-yl)methyl 2-fluorobenzenesulfonate* (**4a-8**). White solid; m.p.: 51–52 °C; yield, 82.5%; ^1^H NMR (400 MHz, Acetone) δ 7.97–7.87 (m, 2H, Ar-H), 7.52–7.46 (m, 2H, Ar-H), 5.52 (s, 2H, -CH_2_-), 2.71 (s, 3H, -CH_3_). ^13^C NMR (100 MHz, Acetone) δ: 166.87, 161.12, 159.11 (d, *J* = 257.6 Hz), 137.54 (d, *J* = 8.8 Hz), 130.94, 125.17 (d, *J* = 3.9 Hz), 123.50 (d, *J* = 13.9 Hz), 117.59 (d, *J* = 20.7 Hz), 61.03, 13.68. ^19^F NMR (376 MHz, Acetone) δ -109.11. HRMS calculated for C_10_H_10_FO_4_N_2_S_2_ [M + H]^+^ 305.00605, found 305.00592.

*(5-(methylthio)-1,3,4-oxadiazol-2-yl)methyl 3-fluorobenzenesulfonate* (**4a-9**). White liquid; yield, 72.5%; ^1^H NMR (400 MHz, CDCl_3_) δ 7.66 (ddd, *J* = 7.9, 1.6, 1.0 Hz, 1H, Ar-H), 7.59–7.54 (m, 1H, Ar-H), 7.51 (td, *J* = 8.1, 5.2 Hz, 1H, Ar-H), 7.34 (tdd, *J* = 8.3, 2.5, 0.9 Hz, 1H, Ar-H), 5.19 (s, 2H, -CH_2_-), 2.63 (s, 3H, -CH_3_). ^13^C NMR (100 MHz, CDCl_3_) δ 167.75, 162.34 (d, *J* = 253.3 Hz), 160.43, 136.89 (d, *J* = 7.2 Hz), 131.38 (d, *J* = 7.8 Hz), 123.94 (d, *J* = 3.5 Hz), 121.87 (d, *J* = 21.1 Hz), 115.51 (d, *J* = 24.9 Hz), 60.01, 14.50. ^19^F NMR (376 MHz, CDCl_3_) δ -108.20. HRMS calculated for C_10_H_10_FO_4_N_2_S_2_ [M + H]^+^ 305.00605, found 305.00571.

*(5-(methylthio)-1,3,4-oxadiazol-2-yl)methyl 3-chlorobenzenesulfonate* (**4a-10**). White liquid; yield, 81.5%; ^1^H NMR (400 MHz, CDCl_3_) δ 7.90 (t, *J* = 1.9 Hz, 1H, Ar-H), 7.84–7.79 (m, 1H, Ar-H), 7.67 (ddd, *J* = 8.1, 2.0, 1.0 Hz, 1H, Ar-H), 7.53 (t, *J* = 8.0 Hz, 1H, Ar-H), 5.27 (s, 2H, -CH_2_-), 2.71 (s, 3H, -CH_3_). ^13^C NMR (100 MHz, CDCl_3_) δ 167.77, 160.41, 136.71, 135.75, 134.65, 130.76, 128.07, 126.18, 60.02, 14.51. HRMS calculated for C_10_H_10_ClO_4_N_2_S_2_ [M + H]^+^ 320.97650, found 320.97635.

*(5-(ethylthio)-1,3,4-oxadiazol-2-yl)methyl 4-fluorobenzenesulfonate* (**4a-11**). White liquid; yield, 80.5%; ^1^H NMR (400 MHz, Acetone-*d_6_*) δ 8.04 (dd, *J* = 9.0, 5.0 Hz, 2H, Ar-H), 7.48 (t, *J* = 8.8 Hz, 2H, Ar-H), 5.45 (s, 2H, -CH_2_-), 3.26 (q, *J* = 7.3 Hz, 2H, -CH_2_-), 1.42 (t, *J* = 7.3 Hz, 3H, -CH_3_). ^13^C NMR (100 MHz, Acetone-*d_6_*) δ 166.09 (d, *J* = 254.8 Hz), 166.04, 161.15, 131.73 (d, *J* = 3.4 Hz), 131.29 (d, *J* = 10.2 Hz), 116.92 (d, *J* = 23.3 Hz), 60.58, 26.54, 14.19. ^19^F NMR (376 MHz, Acetone-*d_6_*) δ -104.41. HRMS calculated for C_11_H_12_O_4_N_2_ClS_2_ [M + H]^+^ 319.02170, found 319.02142.

*(5-(ethylthio)-1,3,4-oxadiazol-2-yl)methyl 4-chlorobenzenesulfonate* (**4a-12**). White liquid; yield, 78.5%; ^1^H NMR (400 MHz, CDCl_3_) δ 7.80–7.75 (m, 2H, Ar-H), 7.50–7.45 (m, 2H, Ar-H), 5.19 (s, 2H, -CH_2_-), 3.17 (q, *J* = 7.4 Hz, 2H, -CH_2_-), 1.40 (t, *J* = 7.4 Hz, 3H, -CH_3_). ^13^C NMR (100 MHz, CDCl_3_) δ 167.14, 160.31, 141.34, 133.66, 129.82, 129.53, 59.96, 26.97, 14.55. HRMS calculated for C_11_H_12_O_4_N_2_ClS_2_ [M + H]^+^ 334.99215, found 334.99191.

*(5-(ethylthio)-1,3,4-oxadiazol-2-yl)methyl 4-bromobenzenesulfonate* (**4a-13**). White liquid; yield, 72.5%; ^1^H NMR (400 MHz, CDCl_3_) δ 7.69 (d, *J* = 8.8 Hz, 2H, Ar-H), 7.63 (d, *J* = 8.8 Hz, 2H, Ar-H), 5.19 (s, 2H, -CH_2_-), 3.17 (q, *J* = 7.4 Hz, 2H, -CH_2_-), 1.40 (t, *J* = 7.4 Hz, 3H, -CH_3_). ^13^C NMR (100 MHz, CDCl_3_) δ 167.13, 160.30, 134.22, 132.80, 129.92, 129.54, 59.98, 26.99, 14.55. HRMS calculated for C_11_H_12_O_4_N_2_BrS_2_ [M + H]^+^ 378.94164, found 378.94110.

*(5-((2-fluoroethyl)thio)-1,3,4-oxadiazol-2-yl)methyl 4-fluorobenzenesulfonate* (**4a-14**). White liquid; yield, 62.5%; ^1^H NMR (400 MHz, Acetone-*d_6_*) δ 8.05 (dd, *J* = 9.0, 5.0 Hz, 2H, Ar-H), 7.48 (t, *J* = 8.8 Hz, 2H, Ar-H), 5.45 (s, 2H, -CH_2_-), 4.81 (t, *J* = 5.8 Hz, 1H, -CH-), 4.69 (t, *J* = 5.8 Hz, 1H, -CH-), 3.64 (t, *J* = 5.8 Hz, 1H, -CH-), 3.59 (t, *J* = 5.8 Hz, 1H, -CH-).^13^C NMR (100 MHz, Acetone-*d_6_*) δ 166.12 (d, *J* = 254.9 Hz), 165.41, 161.48, 131.69 (d, *J* = 3.2 Hz), 131.32 (d, *J* = 10.1 Hz), 116.93 (d, *J* = 23.2 Hz), 80.99 (d, *J* = 169.1 Hz), 60.50, 32.06 (d, *J* = 22.0 Hz). ^19^F NMR (376 MHz, Acetone-*d_6_*) δ -104.33, -216.98. HRMS calculated for C_11_H_11_O_4_N_2_F_2_S_2_ [M + H]^+^ 337.01228, found 337.01169.

*(5-((2-fluoroethyl)thio)-1,3,4-oxadiazol-2-yl)methyl 4-chlorobenzenesulfonate* (**4a-15**). White liquid; yield, 80.5%; ^1^H NMR (400 MHz, CDCl_3_) δ 7.81–7.75 (m, 2H, Ar-H), 7.50–7.45 (m, 2H, Ar-H), 5.19 (s, 2H, -CH_2_-), 4.72 (t, *J* = 5.7 Hz, 1H, -CH-), 4.60 (t, *J* = 5.7 Hz, 1H, -CH-), 3.49 (t, *J* = 5.7 Hz, 1H, -CH-), 3.43 (t, *J* = 5.7 Hz, 1H, -CH-). ^13^C NMR (100 MHz, CDCl_3_) δ 166.17, 160.81, 141.37, 133.62, 129.81, 129.51, 80.62 (d, *J* = 172.1 Hz), 59.74, 32.33 (d, *J* = 22.2 Hz). ^19^F NMR (376 MHz, CDCl_3_) δ -215.71. HRMS calculated for C_11_H_11_O_4_N_2_ClFS_2_ [M + H]^+^ 352.98273, found 352.98209.

*(5-((2-fluoroethyl)thio)-1,3,4-oxadiazol-2-yl)methyl 4-bromobenzenesulfonate* (**4a-16**). White solid; m.p.: 90–91 °C; yield, 80.5%; ^1^H NMR (400 MHz, CDCl_3_) δ 7.73–7.68 (m, 2H, Ar-H), 7.68–7.61 (m, 2H, Ar-H), 5.19 (s, 2H, -CH_2_-), 4.72 (t, *J* = 5.7 Hz, 1H, -CH-), 4.61 (t, *J* = 5.7 Hz, 1H, -CH-), 3.49 (t, *J* = 5.7 Hz, 1H, -CH-), 3.44 (t, *J* = 5.7 Hz, 1H, -CH-); ^13^C NMR (100 MHz, CDCl_3_) δ 166.21, 160.80, 134.20, 132.83, 129.98, 129.55, 80.66 (d, *J* = 172.3 Hz), 59.78, 32.37 (d, *J* = 22.2 Hz). ^19^F NMR (376 MHz, CDCl_3_) δ -215.68. HRMS calculated for C_11_H_11_O_4_N_2_BrFS_2_ [M + H]^+^ 396.93222, found 396.93161.

*(5-(methylthio)-1,3,4-oxadiazol-2-yl)methyl dimethylsulfamate* (**4a-17**). White liquid; yield, 80.5%; ^1^H NMR (400 MHz, CDCl_3_) δ 5.27 (s, 2H, -CH_2_-), 2.92 (s, 6H, -N(CH_3_)_2_), 2.76 (s, 3H, -CH_3_). ^13^C NMR (100 MHz, CDCl_3_) δ 167.55, 161.45, 59.55, 38.46, 14.56. HRMS calculated for C_6_H_12_O_4_N_3_S_2_ [M + H]^+^ 254.02637, found 254.02617.

*(5-(methylthio)-1,3,4-oxadiazol-2-yl)methyl 2-(trifluoromethyl)benzenesulfonate* (**4a-18**). White liquid; yield, 64.5%; ^1^H NMR (400 MHz, CDCl_3_) δ 8.20 (d, *J* = 7.5 Hz, 1H, Ar-H), 7.88 (d, *J* = 7.5 Hz, 1H, Ar-H), 7.73 (ddd, *J* = 14.0, 11.1, 6.7 Hz, 2H, Ar-H), 5.26 (s, 2H, -CH_2_-), 2.63 (s, 3H, -CH_3_). ^13^C NMR (100 MHz, CDCl_3_) δ 167.77, 160.48, 134.50, 133.84, 132.58, 132.35, 128.84 (d, *J* = 6.1 Hz), 123.47, 120.75, 60.04, 14.48. ^19^F NMR (376 MHz, CDCl_3_) δ -58.49. HRMS calculated for C_11_H_10_O_4_N_2_F_3_S_2_ [M + H]^+^ 355.00286, found 355.00241.

*(5-(methylthio)-1,3,4-oxadiazol-2-yl)methyl benzoate* (**5a-1**). White solid; m.p.: 30–31 °C; yield, 77.5%; ^1^H NMR (400 MHz, CDCl_3_) δ 8.09 (d, *J* = 7.1 Hz, 2H, Ar-H), 7.62 (t, *J* = 7.4 Hz, 1H, Ar-H), 7.48 (t, *J* = 7.7 Hz, 2H, Ar-H), 5.52 (s, 2H, -CH_2_-), 2.76 (s, 3H, -CH_3_). ^13^C NMR (100 MHz, CDCl_3_) δ 166.92, 165.49, 162.81, 133.74, 130.00, 128.70, 128.58, 55.53, 14.60. HRMS calculated for C_11_H_11_O_3_N_2_S [M + H]^+^ 251.04849, found 251.04831.

*(5-(methylthio)-1,3,4-oxadiazol-2-yl)methyl 4-fluorobenzoate* (**5a-2**). White solid; m.p.: 53–54 °C; yield, 75.5%; ^1^H NMR (400 MHz, CDCl_3_) δ 8.02 (dd, *J* = 9.0, 5.4 Hz, 2H, Ar-H), 7.06 (t, *J* = 8.7 Hz, 2H, Ar-H), 5.42 (s, 2H, -CH_2_-), 2.67 (s, 3H, -CH_3_). ^13^C NMR (100 MHz, CDCl_3_) δ 166.99, 166.22 (d, *J* = 255.3 Hz), 164.52, 162.68, 132.66 (d, *J* = 9.5 Hz), 124.94 (d, *J* = 3.0 Hz), 115.85 (d, *J* = 22.1 Hz), 55.62, 14.61. ^19^F NMR (376 MHz, CDCl_3_) δ -104.05. HRMS calculated for C_11_H_10_O_3_N_2_FS [M + H]^+^ 269.03907, found 269.03897.

*(5-(methylthio)-1,3,4-oxadiazol-2-yl)methyl 4-chlorobenzoate* (**5a-3**). White solid; m.p.: 53–54 °C; yield, 74.5%; ^1^H NMR (400 MHz, CDCl_3_) δ 7.95–7.90 (m, 2H, Ar-H), 7.39–7.34 (m, 2H, Ar-H), 5.42 (s, 2H, -CH_2_-), 2.67 (s, 3H, -CH_3_), ^13^C NMR (100 MHz, CDCl_3_) δ 166.98, 164.64, 162.60, 140.32, 131.37, 128.97, 127.15, 55.70, 14.61. HRMS calculated for C_11_H_10_O_3_N_2_ClS [M + H]^+^ 285.00952, found 285.00958.

*(5-(methylthio)-1,3,4-oxadiazol-2-yl)methyl 4-bromobenzoate* (**5a-4**). White solid; m.p.: 82–83 °C; yield, 70.5%; ^1^H NMR (400 MHz, CDCl_3_) δ 7.86 (d, *J* = 8.7 Hz, 2H, Ar-H), 7.54 (d, *J* = 8.7 Hz, 2H, Ar-H), 5.42 (s, 2H, -CH_2_-), 2.68 (s, 3H, -CH_3_). ^13^C NMR (100 MHz, CDCl_3_) δ 167.01, 164.81, 162.57, 131.98, 131.48, 129.06, 127.59, 55.70, 14.61. HRMS calculated for C_11_H_10_O_3_N_2_BrS [M + H]^+^ 328.95900, found 328.95895.

*(5-(methylthio)-1,3,4-oxadiazol-2-yl)methyl 4-methoxybenzoate* (**5a-5**). White solid; m.p.: 35–36 °C; yield, 79.5%; ^1^H NMR (400 MHz, CDCl_3_) δ 8.04 (d, *J* = 9.0 Hz, 2H, Ar-H), 6.95 (d, *J* = 9.0 Hz, 2H, Ar-H), 5.49 (s, 2H, -CH_2_-), 3.89 (s, 3H, -CH_3_), 2.76 (s, 3H, -CH_3_).^13^C NMR (100 MHz, CDCl_3_) δ 166.83, 165.18, 163.97, 163.02, 132.13, 121.01, 113.84, 55.52, 55.28, 14.60. HRMS calculated for C_12_H_13_O_4_N_2_S [M + H]^+^ 281.05905, found 281.05884.

*(5-(methylthio)-1,3,4-oxadiazol-2-yl)methyl dimethylcarbamate* (**5a-6**). White liquid; yield, 81.5%; ^1^H NMR (400 MHz, CDCl_3_) δ 5.28 (s, 2H, -CH_2_-), 2.97 (s, 3H, -CH_3_), 2.95 (s, 3H, -CH_3_), 2.75 (s, 3H, -CH_3_). ^13^C NMR (100 MHz, CDCl_3_) δ 166.51, 163.49, 155.10, 56.13, 36.79, 36.05, 14.60. HRMS calculated for C_7_H_12_O_3_N_3_S [M + H]^+^ 218.05939, found 218.05922.

### 3.3. X-ray Diffraction Analysis

All target compounds had been confirmed by ^1^H NMR, ^13^C NMR and high-resolution mass spectrometry (HRMS). After a preliminary in vitro and in vivo bactericidal analysis, Compound **4a-2** had the best bactericidal activity. The structural composition of compound **4a-2** was determined by single crystal X-ray analysis.

Crystal structure of compound **4a-2** (C_10_H_9_FO_4_N_2_S_2_) is shown in Figure 5. Colorless crystal of compound **4a-2** (0.4 × 0.28 × 0.2 mm) is monoclinic system and space group C 2/C. Cell parameters: a = 26.431(2), b = 5.1560(5), c = 21.7311(18), alpha = 90, beta = 121.147(4), gamma = 90, V = 2534.5(4), Z = 8. Cell dimensions and intensities were measured at 298 K on Bruker SMART diffractometer with MoK\a radiation (λ = 0.71073 Å). A total of 2215 reflections were measured, of which 1662 were unique in the range of 3.10 <θ< 25.02^°^ (h, -18 to 31; k, -6 to 6; l, -25 to 24), The structure was solved by direct method with the SHELXL-2014 program. All of the non-H atoms were refined anisotropically by full-matrix least-squares to give the final R = 0.0409 and WR2 = 0.1075. All hydrogen atoms were computed and refined using a riding model. The completeness of the crystal data is 99.4%. The atomic coordinates for **4a-2** have been deposited at the Cambridge Crystallographic Data Centre. CCDC 1,975,227 contains the Appendix A for this paper.

### 3.4. Antibacterial Bioassay by Scanning Electron Microscopy

The sample preparation method was as follows [30]. A certain quantity of microcentrifuge tube (2 ml) was prepared and added bacteria solution of *Xoo* (1.5 mL). Then microcentrifuge tubes were washed with PBS buffer and centrifuged 3 times, in order to discard supernatant, microcentrifuge tubes were centrifuge at 7000 rpm for one minute. Compound **4a-2** was added into the centrifuge tube to prepare 0 µg/mL, 25 µg/mL and 50 µg/mL solution, three parallel groups for each concentration. 2.5% glutaraldehyde fixing solution was added to each microcentrifuge tube for 12 h and removed. Next, microcentrifuge tubes were washed by 30%, 50%, 70%, 90% and absolute ethanol in this order. At last, the samples were flattened and sprayed gold (45s) for observing by SEM.

### 3.5. Antibacterial Bioassay In Vitro

The inhibitory efficiency of target compounds on two bacteria in vitro was tested by the turbidimeter method at different concentrations [10]. For initial screening of all 24 compounds, the solution concentration was set at 200 and 100 μg/mL which was incubated with bacterial solution and then procedurally measured for the OD value. A solution with no compound was set as a negative check and bismerthiazol and thiodiazole copper served as the positive control. Compounds that were active at this concentration were further tested at five lower gradient concentrations to get EC_50_. Data were collected in triplicate for each compound concentration. Based on the OD value, the inhibitory effect of the compound on bacteria was calculated. I (%) = (CK-T)/T × 100%. I (%) meant inhibition rate. CK meant the OD value of non-drug control group. T meant the OD value of drug group.

### 3.6. Antibacterial Activity Bioassay In Vivo

Compounds **4a**-**1**, **4a**-**2** and **4a**-**3** were tested for the protective and curative activity in vivo against rice bacterial leaf blight by leaf-cutting method [10,30] at 200 µg/mL, with comparing to bismerthiazol and thiodiazole copper. A negative control check (CK) was set up identically with absence of the test compound. Data were collected in triplicate treatment. Then the control efficiency could be calculated by analyzing plant disease index. Control efficiency (%) = (C − T)/C × 100%, where C represented the plant disease index of the negative control CK; T represented the disease index of plant with the compound treatment.

## 4. Conclusions

In summary, 24 novel sulfonate/carboxylate functionalized 1,3,4-oxadiazole derivatives were synthesized and evaluated for antibacterial activity on both bacteria *Xanthomonas oryzae* pv. *oryzae* and *Xanthomonas axonopodis* pv. *citri*. Among them, ten compounds (**4a-1 to 4a-4 and 4a-11 to 4a-16**) showed extensive potency on the *Xoo* in vitro. Four (**4a-1 to 4a-4)** of them also performed well on *Xac* in vitro. In particular, compound **4a-2** with the best antibacterial activity in vitro indicated excellent protective and curative activity against rice bacterial leaf blight in vivo. Furtherly, scanning electron microscope analysis on **4a-2** verified its antibacterial action mechanism. Structure-activity relationship illustrated that sulfonate structure (**4a**), rather than carboxylate moiety(**5a**), play important role for inhibitory effect of target compounds. In conclusion, as expected, 1,3,4-Oxadiazole derivatives containing sulfonate moiety showed promise antibacterial activity and might provide potent plant bactericide.

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
