# Peer review of "Synthesis and Antibacterial Evaluation of Novel 1,3,4-Oxadiazole Derivatives Containing Sulfonate/Carboxylate Moiety"

_molecules, 2020, doi:10.3390/molecules25071488_

Round 1
Reviewer 1 Report
The ms of Zhan/Jin at al describes the synthesis of ne sulfonate or carboxylate-functionalized 1,3,4-oxadiazole derivatives. As such, the ms is sound, however, more chemistry would be required, and the bioactivity part should be concentrated.
The following other shortcomings should be eliminated:
- The numbering of the compounds (1,2,3,4 I and 4II) is rather awkward! Pls renumber applying 4a- and 5a- instead of 4I and 4II, respectively.
- Presently, there is no spectral characterization of the new oxadiazoles in the main body of the ms. This is a serious defficiency!
- A few times “positive control” is mentioned. What is this?
- The usage of English is rather poor throughout the ms. The ms should be thoroughly refined by a native English speaker. Just a few examples:
- Our previous work has found”, “explore to synthesize”, “Antibacterial activities” use singular), “ HRMS (ESI). And 19F “ antibacterial activities were” (use singular. “potency of tested compounds“ (insert”the”), “acted best for combating”, “with established outstanding “(pls delete”establishd”), “ability to visually inspect”, “4I and 4Ii series derivatives”,. “sulfonate was very important”(the sulfonate structure was very important
In summary a major revision is recommended before reconsideration.
Author Response
Response to Reviewers 1
Reviewer 1:
- The ms of Zhan/Jin at al describes the synthesis of ne sulfonate or carboxylate-functionalized 1,3,4-oxadiazole derivatives. As such, the ms is sound, however, more chemistry would be required, and the bioactivity part should be concentrated.
Ans.: Thanks and the manuscript has been revised according to your suggestion.
Firstly, according to your comments, we describe more chemistry information into manuscript, including the general introduction about the synthesis procedure and targeted chemicals spectral characterization data added in 2.1 and 3.2.4 section respectively.
Revision in 2.1 is “As described in Scheme 1, starting from ethyl glycolate, the key intermediate (5-mercapto-1,3,4-oxadiazol-2-yl) methanol 2 was synthesized in two steps involving acylation and cyclization. Subsequently, intermediate 2 was converted into its corresponding thioether derivative 3 by thioetherification with R1I. Finally, the target compounds 4a/5a was obtained by esterification with R2SOOCl/R3COCl. The structures of all target compounds were confirmed by nuclear magnetic resonance spectra including 1H NMR, 13C NMR and electrospray ionization high-resolution mass spectrometry (ESI-HRMS). And fluorine nuclear magnetic resonance (19F NMR) was involved for some fluoride structures.
Revision in 3.2.4. is the supplement spectral characterization data as in the answer to your third question.
Secondly, the bioactivity method part in 3.5 and 3.6 section had been concentrated in by rewriting, the specific modification content are as followings,
The paragraph in section 3.5 Antibacterial Bioassay In Vitro from line 35 “First, target compound was dissolved by DMSO (300 µL) to prepare drug solution. Different volumes of drug solution were taken into 15 mL test tubes by a pipette, and then diluted to 4 mL with 0.1% Tween-20 sterile distilled water. Finally, 200 µg/mL, 100 µg/mL, 50 µg/mL, 25 µg/mL, and 12.5 µg/mL working concentration were prepared by mixing 4 mL nutrient broth medium (NB). At last, test tubes were added bacterial solution (40 µL), and cultured in a shaker at 180 rpm and 28℃. Each concentration was repeated three times for each compound. When the OD value of the control group CK (without drug) was in the logarithmic growth phase of 0.6-0.8, the OD value of all tubes could be measured at 595 nm.” was changed to “For initial screening of all 24 compounds, the solution concentration was set at 200 and 100 μg/mL which was incubated with bacterial solution and then procedurally measured for the OD value. Solution with no compound was set as negative check and Bismerthiazol, Thiodiazole copper served as positive control. Compounds that were active at this concentration were further tested at five lower gradient concentrations to get EC50. Data were collected in triplicate for each compound concentration.”
Section 3.6 “Antibacterial Activity Bioassay In Vivo”: Original: “The compounds 4I-1, 4I-2 and 4I-3 were tested for the protective and curative activity in vivo against rice bacterial leaf blight according to the following methods [33]. Before the test, compounds 4I-1, 4I-2, 4I-3 and the negative control Bismerthiazol, Thiodiazole copper was diluted with 0.1% Tween-20 water to configure agents (200µg/mL), respectively. The protection bioassay was as follows. On the first day, the formulated drugs 4I-1, 4I-2, 4I-3, Biserthiazol and Thiodiazole copper were sprayed evenly on rice leaves. The next day, cut off the 2-3 cm part of the rice leaf tip, soaked the wound part in the bacterial solution for one minute, and repeated the soaking process three times. After 14 days, the protective effect of the agent on rice was evaluated. The test method for curative activities was similarly. The method for measuring the control efficiency and the disease index were as follows, Control efficiency (%) = (C − T)/C × 100%, where C represented the disease index of the positive control CK; T represented the disease index of the rice treated with the agent.” was revised as: “Compounds 4a-1, 4a-2 and 4a-3 were tested for the protective and curative activity in vivo against rice bacterial leaf blight by leaf-cutting method [10] at 200µg/mL, with comparing to Bismerthiazol, Thiodiazole copper. A negative control check (CK) was set up identically with absence of the test compound. Data were collected in triplicate treatment. Then the control efficiency could be calculated by analyzing plant disease index. Control efficiency (%) = (C − T)/C × 100%, where C represented the plant disease index of the negative control CK; T represented the disease index of plant with the compound treatment.”
2) The numbering of the compounds (1,2,3,4 I and 4II) is rather awkward! Pls renumber applying 4a- and 5a- instead of 4I and 4II, respectively.
Ans.: Thanks. And according to your suggestions we revised the numbering of the compounds into 1, 2, 3, 4a and 5a in our manuscript. And all the numbering was adjusted wherever it appears both in the MS and Supporting Information file.
3)Presently, there is no spectral characterization of the new oxadiazoles in the main body of the ms. This is a serious deficiency!
Ans.: Thanks for your kind and accurate commentary. Sure it is a deficiency without spectral information. Since we had originally put all this spectral data in the Supporting Information and submitted supplementary file. According to your comment, we set all spectral data of target compounds in the Experimental part “3.2.4 Preparation of Target compound” of the MS. The following is an example of 4a-1 for all the compounds.
(5-(methylthio)-1,3,4-oxadiazol-2-yl)methyl benzenesulfonate (4a-1). White solid; m.p.: 54-55 °C; yield, 80.5%; 1H NMR (400 MHz, CDCl3) δ 7.93 (dd, J = 8.4, 1.2 Hz, 2H, Ar-H), 7.70 (t, J = 7.5 Hz, 1H, Ar-H), 7.58 (t, J = 7.8 Hz, 2H, Ar-H), 5.23 (s, 2H, -CH2-), 2.69 (s, 3H, -CH3). 13C NMR (100 MHz, CDCl3) δ 167.60, 160.69, 135.10, 134.50, 129.49, 128.13, 59.75, 14.51. HRMS calculated for C10H11O4N2S2 [M + H]+ 287.01547, found 287.01529.
4) A few times “positive control” is mentioned. What is this?
Ans.: To make it clear, we necessarily listed the name Bismerthiazole and Thiodiazole-copper instead of “positive control” in the text.
For more information, “positive control” here is referring to commercial bactericides. As we described in “2.1 chemistry” Bismerthiazole and Thiodiazole-copper served as positive controls to compare the bactericidal potency of tested compounds.” And commercially available Bismerthiazol and Thiodiazole-copper are labeled for the treat of disease including what we tend to evaluate. Their structure information are as following: a) Bismerthiazol: 5,5'-(methylenediimino)bis-1,3,4-Thiadiazole-2(3H)-thione); b) Thiodiazole-copper: 2-amino-5-mercapto-1,3,4-thiadiazole-copper).
5) The usage of English is rather poor throughout the ms. The ms should be thoroughly refined by a native English speaker. Just a few examples: Our previous work has found”, “explore to synthesize”, “Antibacterial activities” (use singular), “HRMS (ESI)”. And 19F “ antibacterial activities were” (use singular). “potency of tested compounds” (insert “the”), “acted best for combating”, “with established outstanding” (pls delete “established”), “ability to visually inspect”, “4I and 4Ii series derivatives”, “sulfonate was very important” (the sulfonate structure was very important).
Ans.: We appreciated so much for all these detailed suggestions and we take all the suggestion and correct the error one by one. In addition, as suggested by honorable reviewer, we had further polished to correct all typos, grammatical error and punctuations and improper wordings, besides the partial list given below.
Line 45 “Our previous work has found” was changed as “We previously found”.
Line 53 “explore to synthesize” in Introduction section was deleted by rewriting the whole paragraph.
Original: Based on the previous research on 1,3,4-oxadiazole, the splicing of oxymethyl and 1,3,4-oxadiazole has excellent activity [10,30]. Now we explore to synthesize a new series of compounds by introducing sulfonate or carboxylate to 1,3,4-oxadiazole and expected to discover new structures with potential antibacterial activities. Revision: “In addition, we reported the splicing of oxymethyl and 1,3,4-oxadiazole sulfone derivatives could provide excellent antibacterial activity [10,30]. Based on those prior work, we hypothesized that sulfonate/carboxylate moiety functionalized 1,3,4-oxadiazole derivatives might also show promise antibacterial activity. Hence in the present work, a series of novel compounds were synthesized by introducing sulfonate or carboxylate moiety to 1,3,4-oxadiazole to discover new structures with potential antibacterial activity.”
All included “antibacterial activities” was changed as “antibacterial activity”. And for consistent, all other “activities” were also change to “activity” singular, and the “activities were” to “activity was”.
Line 72, “HRMS (ESI)” was changed as “electrospray ionization high-resolution mass spectrometry(ESI-HRMS)when first mentioned, which was following abbreviated as HRMS.
Line 74 “19F NMR” sentence was changed to “Fluorine nuclear magnetic resonance 19F NMR was involved for some fluoride structures”.
Line 83 “the bactericidal potency of tested compounds” was changed as” “the bactericidal potency of the tested compounds” as you mentioned.
Line 102 “acted best for combating” was changed as “performed the best on”.
Line 117 “with established outstanding” was changed as “with outstanding”.
Line 139 “ability to visually inspect” was changed as “ability to observe”.
Line 153 “the 4I and 4II series derivatives” was changed as “those 4a and 5a derivatives”. (All the other 4I and 4II were all taken care of.)
Line 157 “sulfonate was very important” was change to “the sulfonate structure was very important”
Line 371 “test tubes were” was omitted by rewriting the corresponding section.
6) In summary a major revision is recommended before reconsideration.
Ans.: As suggested by the honorable reviewer, the manuscript has been thoroughly refined by a native English speaker to correct all typos, grammatical error and punctuations and improper wordings. As mentioned above, several sections was revised and rewritten.
The second paragraph in section “2.5 Structure-Activity Relationship (SAR) Analyses”was revised.
Original: Further antibacterial tests on Xoo and Xac showed that 4-substituted halogenated phenyl sulfonate derivatives expressed significant antibacterial activity which appeared to decrease with increase in the size of the halogen substituent. Therefore, the order follows the antibacterial trend 4I-2 (F) > 4I-3 (Cl) > 4I-4 (Br), indicating that 4-F substituted derivatives was most advantageous in conferring antibacterial activity. In this regard, it was consistent with previous reports [10]. Revision: Further antibacterial evaluation on Xoo and Xac showed that 4-substituted halogenated phenyl sulfonate derivatives expressed significant antibacterial activity. Three (4a-2, 4a-3, 4a-4) of them worked well on both Xoo and Xac, which appeared an obvious decreasing potency(50.10, 87.23, 99.40 µmol/L) with increasing halogen size in 4a-2(R2=F), 4a-3(R2=Cl), 4a-4(R2=Br) respectively. In this regard, it was consistent with previous reports [10]. The other six compounds 4a-11-4a-16 (R1=C2H5) also showed extensive potency on the Xoo. However, their EC50 are slightly decreased like 98.02, 95.31, 86.38 µmol/L and not necessarily following the tendency 4a-11(F) >4a-12(Cl) >4a-13(Br). It can be refereed that R1 in thioether side chain also make difference in the activity of the structure. So in particular, when R1=CH3, R2=F, compound 4a-2 would be the most promising compound both in vitro and in vivo against the tested plant bacteria.
The final Conclusion section also revised.
Original: “In summary, we have synthesized 24 novel 1,3,4-oxadiazole derivatives containing sulfonate/ carboxylate structure. The evaluation of the inhibition efficiency of compounds 4I-1-4I-3 on both bacteria Xanthomonas oryzae pv. oryzae and Xanthomonas axonopodis pv. citri were better than the commercial agriculture drugs Bismerthiazol and Thiodiazole copper, meanwhile, the evaluation of EC50 of 4I-1-4I-3 on Xoo and Xac were more excellent than the both drugs. Among them, 4I-2 had the best antibacterial activity in vitro. In addition, compound 4I-2 had excellent in vivo protective and curative activities against rice bacterial leaf blight. Next, the scanning electron microscope verified the antibacterial effect of the compound 4I-2. By studying the structure-activity relationship of compound 4I-2, It was a commercial pesticide that compound 4I-2 had the potential to be, and it could be further studied.” Revision: “In summary, 24 novel sulfonate/carboxylate functionalized 1,3,4-oxadiazole derivatives were synthesized and evaluated for antibacterial activity on both bacteria Xanthomonas oryzae pv. oryzae and Xanthomonas axonopodis pv. citri. Among them, ten compounds (4a-1 to 4a-4 and 4a-11 to 4a-16) showed extensive potency on the Xoo in vitro. Four (4a-1 to 4a-4) of them also performed well on Xac in vitro. In particular, compound 4a-2 with the best antibacterial activity in vitro indicated excellent protective and curative activity against rice bacterial leaf blight in vivo. Furtherly, scanning electron microscope (SEM) analysis on 4a-2 verified its antibacterial action mechanism. Structure-activity relationship illustrated that sulfonate structure(4a), rather than carboxylate moiety(5a), play important role for inhibitory effect of target compounds. In conclusion, as expected, 1,3,4-Oxadiazole derivatives containing sulfonate moiety showed promise antibacterial activity and might provide potent plant bactericide.”
X-ray Diffraction Analysis were also revise by updating the processing version.
Line 339 Insert the sentence “and space group C 2/C. Cell parameters: a=26.431(2), b=5.1560(5), c=21.7311(18), alpha=90, beta=121.147(4), gamma=90, V=2534.5(4), Z=8.”
Line 346 Insert the sentence “The completeness of the crystal data is 99.4%.”

Reviewer 2 Report
The manuscript is completely missing the chemistry experimental part and results of the synthesis are not supported by any data.
Discussion of structure-activity without IC50 values in molar concentrations is mainly meaningless. The conclusion that antibacterial activity "appeared to decrease with increase in the size of the halogen substituent" could be wrong just because of that. Increase of the halogen size contribute to the molecular mass making molar concentration lower. Maybe this lower concentration rather than difference in the structure plays role. The precise answer can be given only on the basis of the comparisons of IC50 values in molar concentration.
Author Response
Response to Reviewer 2
- 1) The manuscript is completely missing the chemistry experimental part and results of the synthesis are not supported by any data.
Ans.: Thanks for your comments and we appreciate that you pointed out the “missing” part of our MS. we had originally put all this spectral characterization data and spectrum graph in the “Supporting Information” and submitted as supplementary file, which make it a serious deficiency for the MS. According to your recommendation, we added the relevant content into 2.1 chapter and 3.2 chapter. The major content is below.
In section 2.1 chemistry “As described in Scheme 1, starting from ethyl glycolate, the key intermediate (5-mercapto-1,3,4-oxadiazol-2-yl) methanol 2 was synthesized in two steps involving acylation and cyclization. Subsequently, intermediate 2 was converted into its corresponding thioether derivative 3 by thioetherification with R1I. Finally, the target compounds 4a/5a was obtained by esterification with R2SOOCl/R3COCl. The structures of all target compounds were confirmed by nuclear magnetic resonance spectra including 1H NMR, 13C NMR and electrospray ionization high-resolution mass spectrometry (ESI-HRMS). And fluorine nuclear magnetic resonance (19F NMR) was involved for some fluoride structures.”
The following is example 4a-1. And all the compounds were also listed in same style.
(5-(methylthio)-1,3,4-oxadiazol-2-yl)methyl benzenesulfonate (4a-1). White solid; m.p.: 54-55 °C; yield, 80.5%; 1H NMR (400 MHz, CDCl3) δ 7.93 (dd, J = 8.4, 1.2 Hz, 2H, Ar-H), 7.70 (t, J = 7.5 Hz, 1H, Ar-H), 7.58 (t, J = 7.8 Hz, 2H, Ar-H), 5.23 (s, 2H, -CH2-), 2.69 (s, 3H, -CH3). 13C NMR (100 MHz, CDCl3) δ 167.60, 160.69, 135.10, 134.50, 129.49, 128.13, 59.75, 14.51. HRMS calculated for C10H11O4N2S2 [M + H]+ 287.01547, found 287.01529.
Furthermore, the manuscript has been thoroughly refined by a native English speaker to correct all typos, grammatical error and punctuations and improper wordings. And several sections including paraph in Introduction, section 2.5Structure-Activity Relationship (SAR) Analyses, section 3.5, Antibacterial Bioassay In Vitro, section 3.6 Antibacterial Activity Bioassay In Vivo and section 4.Conclusions.
- 2) Discussion of structure-activity without IC50 values in molar concentrations is mainly meaningless. The conclusion that antibacterial activity "appeared to decrease with increase in the size of the halogen substituent" could be wrong just because of that. Increase of the halogen size contribute to the molecular mass making molar concentration lower. Maybe this lower concentration rather than difference in the structure plays role. The precise answer can be given only on the basis of the comparisons of IC50 values in molar concentration.
Ans.: Thanks for the thoughtful suggestion. And we are fully agreed with the honorable reviewer’s points.
We have converted the EC50 mass concentration in µg/mL to molar concentration in µmol/L (by calculated as 1000*C/M, C is the EC50 value in mass, and M is the corresponding molecular weight), and with that change, the regression equation was adjusted correspondingly. It can be seen, your advice and estimation were very reasonable. When the molecular mass was considered, the EC50 trend is generally same for most compounds, but except for 4a-11 4a-12,4a-13. For 4a-11 4a-12,4a-13, where R1=C2H5, R2=F, Cl, Br respectively, their EC50 in mass concentration is 31.27, 31.93, 32.74µg/mL, showing in same level or slightly increase, but their EC50 in molar concentration is 98.02 95.31, 86.38 µmol/L changing in opposite direction with slight decrease.
As the honorable reviewer said “Increase of the halogen size contribute to the molecular mass making molar concentration lower. Maybe this lower concentration rather than difference in the structure plays role”. So our early conclusion "appeared to decrease with increase in the size of the halogen substituent" should be adjusted. At this point, we make revisions of the whole second paragraph in “2.5 Structure-Activity Relationship (SAR) Analyses” section as following:
“Further antibacterial evaluation on Xoo and Xac showed that 4-substituted halogenated phenyl sulfonate derivatives expressed significant antibacterial activity. Three (4a-2, 4a-3, 4a-4) of them worked well on both Xoo and Xac, which appeared an obvious decreasing potency(50.10, 87.23, 99.40 µmol/L ) with increasing halogen size in 4a-2(R2=F), 4a-3(R2=Cl), 4a-4(R2=Br) respectively. In this regard, it was consistent with previous reports [10].
The other six compounds 4a-11-4a-16 (R1=C2H5) also showed extensive potency on the Xoo. However, their EC50 are slightly decreased like 98.02 95.31, 86.38 µmol/L and not necessarily following the tendency 4a-11(F) >4a-12(Cl) >4a-13(Br). It can be refereed that R1 in thioether side chain also make difference in the activity of the structure. So in particular, when R1=CH3, R2=F, compound 4a-2 would be the most promising compound both in vitro and in vivo against the tested plant bacteria.”

Reviewer 3 Report
Antibacterial activities against two phytopathogens, Xanthomonas oryzae pv. oryzae (Xoo) and Xanthomonas axonopodis pv. citri (Xac), were described in this manusprict.
The referee suggests:
The numeration of the compounds was not clear, it should be changed to follow the results more easily.
The crystallographic analysis was not enough described. The program used was too old (SHELX 97 vs SHELX 17). CIF data was not attainable.
The supporting information was not achievable.
Author Response
Response to Reviewer 3
- 1)The numeration of the compounds was not clear, it should be changed to follow the results more easily.
Ans.: Thanks for your good comment on our manuscript, and your recommendation is appreciated. We have changed the numeration of the compounds 1, 2, 3, 4I and 4II into 1, 2, 3, 4a and 5a which was more clear comprehension for tracking results.
- 2)The crystallographic analysis was not enough described. The program used was too old (SHELX 97 vs SHELX 17). CIF data was not attainable.
Ans.: Thanks for your suggestions, we are sorry for the limited experimental conditions and the SHELX 97 too old. Cif file has been reprocessed the now updated to version SHELX 14. We have added the following description to better express the structure of the crystal.
Crystal structure of compound 4a-2 (C10H10FO4N2S2) is shown in Figure 5. Colorless crystal of compound 4a-2 (0.4mm × 0.28mm × 0.2mm) is monoclinic system and space group C 2/C. Cell parameters: a=26.431(2), b=5.1560(5), c=21.7311(18), alpha=90, beta=121.147(4), gamma=90, V=2534.5(4), Z=8. The completeness of the crystal data is 99.4%. The atomic coordinates for 4a-2 have been deposited at the Cambridge Crystallographic Data Centre. CCDC 1975227 contains the supplementary crystallographic data for this paper.
- 3)The supporting information was not achievable.
Ans.: We are sorry for the supporting information missing happened. It will be resubmitted with corresponding changes like the compound numeration.
Thanks for the suggestion and all the required information has been taken care of.
And for your reference, there is another major revision in the MS. As suggested we convert the EC50 mass concentration into molar concentration (calculated by 1000*C/M, C is the EC50 value in mass concentration, and M is the corresponding molecular weight) and the regression equation was adjusted. And the result and discussion would be given on the basis of the comparisons of EC50 values in molar concentration.
Furthermore, the manuscript has been thoroughly refined by a native English speaker to correct all typos, grammatical error and punctuations and improper wordings. Several sections including paraph in Introduction, section 2.5, section 3.5, section 3.6, and section 4.Conclusions”

Round 2
Reviewer 1 Report
This revised ms is accptable for Molecules. Au-s carried out a careful revision on the problems outlined by me.
Author Response
Reviewer 1:
This revised ms is acceptable for Molecules. Au-s carried out a careful revision on the problems outlined by me.
Answer: We appreciated for your dedicated work and kind help on our manuscript.
And there was still a few revisions as listed below for your reference.
- replacing "μmol/L" with more typical "μM",
- reducing significant figure of value % or EC50 to 3 numbers. And remove the original table 1 in supplementary file and using a concise table 1 without regression equation and r with an referring line “c The corresponding regression equation and r value for this EC50 were provided in supplementary data” in the bottom of the table illustration.
- modification in the structure characterization(for fluorinated compounds, coupling constant of J C F coupling indicated in their 13C NMR spectra were characterized)

Reviewer 2 Report
In general, the manuscript has been improved. However, there a numerous issue with style and grammar. An extensive English correction seems to be needed. Addition of experimental data for compounds and copies on NMR spectra as the Supplementary Data would improve credibility of the manuscript. The NMR spectra look quite reasonable and correspond to the proposed structures, but their description in the experimental part (especially for fluorinated compounds) appears incorrect and does not correspond to the provided copies of spectra. Please check.
Please replace "μmol/L" with more typical "μM".
Please remove the abbreviation "SEM" from keywords and explain it only once (the first time mentioned) in the text.
Please reduce number of significant figure in the reported results according to the experiment accuracy. For example, it is definitely incorrect to use 4 (and more!) significant figures for EC50 values or % in biological results. It is also unnecessary to report in the manuscript regression equation and r. They can find a better place in the Supplementary Data.
I think that if authors continue improving their manuscript, it might become publishable.
Author Response
Reviewer 2:
- In general, the manuscript has been improved. However, there a numerous issue with style and grammar. An extensive English correction seems to be needed. Addition of experimental data for compounds and copies on NMR spectra as the Supplementary Data would improve credibility of the manuscript. The NMR spectra look quite reasonable and correspond to the proposed structures, but their description in the experimental part (especially for fluorinated compounds) appears incorrect and does not correspond to the provided copies of spectra. Please check.
Answer: Thank you so much for your dedicated work and kind help on our manuscript.
We appreciate that you pointed out the weakness of our manuscript to help enhance the quality of our work. We make sure this time the description corrected and correspond to spectra. Especially, for fluorinated compounds, coupling constant of J C F coupling indicated in their 13C NMR spectra were characterized in the file as following style.
(5-(methylthio)-1,3,4-oxadiazol-2-yl)methyl 4-fluorobenzenesulfonate (4a-2). White solid; m.p.: 64-65 °C; yield, 86.5%; 1H NMR (400 MHz, Acetone) δ 8.04 (dd, J = 9.0, 5.0 Hz, 2H, Ar-H), 7.48 (t, J = 8.8 Hz, 2H, Ar-H), 5.44 (s, 2H, -CH2-), 2.71 (s, 3H, -CH3). 13C NMR (100 MHz, Acetone) δ 166.81, 166.10 (d, J = 254.9 Hz), 161.21, 131.72 (d, J = 3.5 Hz), 131.30 (d, J = 10.2 Hz), 116.91 (d, J = 23.3 Hz), 60.57, 13.67. 19F NMR (376 MHz, Acetone) δ -104.45. HRMS calculated for C10H10FO4N2S2 [M + H]+ 305.00605, found 305.00592.
(5-(ethylthio)-1,3,4-oxadiazol-2-yl)methyl 4-fluorobenzenesulfonate (4a-11). White liquid; yield, 80.5%; 1H NMR (400 MHz, Acetone-d6) δ 8.04 (dd, J = 9.0, 5.0 Hz, 2H, Ar-H), 7.48 (t, J = 8.8 Hz, 2H, Ar-H), 5.45 (s, 2H, -CH2-), 3.26 (q, J = 7.3 Hz, 2H, -CH2-), 1.42 (t, J = 7.3 Hz, 3H, -CH3). 13C NMR (100 MHz, Acetone-d6) δ 166.09 (d, J = 254.8 Hz), 166.04, 161.15, 131.73 (d, J = 3.4 Hz), 131.29 (d, J = 10.2 Hz), 116.92 (d, J = 23.3 Hz), 60.58, 26.54, 14.19. 19F NMR (376 MHz, Acetone-d6) δ -104.41. HRMS calculated for C11H12O4N2ClS2 [M + H]+ 319.02170, found 319.02142.
(5-(ethylthio)-1,3,4-oxadiazol-2-yl)methyl 4-bromobenzenesulfonate (4a-13). White liquid; yield, 72.5%; 1H NMR (400 MHz, CDCl3) δ 7.69 (d, J = 8.8 Hz, 2H, Ar-H), 7.63 (d, J = 8.8 Hz, 2H, Ar-H), 5.19 (s, 2H, -CH2-), 3.17 (q, J = 7.4 Hz, 2H, -CH2-), 1.40 (t, J = 7.4 Hz, 3H, -CH3). 13C NMR (100 MHz, CDCl3) δ 167.13, 160.30, 134.22, 132.80, 129.92, 129.54, 59.98, 26.99, 14.55. HRMS calculated for C11H12O4N2BrS2 [M + H]+ 378.94164, found 378.94110.
(5-((2-fluoroethyl)thio)-1,3,4-oxadiazol-2-yl)methyl 4-fluorobenzenesulfonate (4a-14). White liquid; yield, 62.5%; 1H NMR (400 MHz, Acetone-d6) δ 8.05 (dd, J = 9.0, 5.0 Hz, 2H, Ar-H), 7.48 (t, J = 8.8 Hz, 2H, Ar-H), 5.45 (s, 2H, -CH2-), 4.81 (t, J = 5.8 Hz, 1H, -CH-), 4.69 (t, J = 5.8 Hz, 1H, -CH-), 3.64 (t, J = 5.8 Hz, 1H, -CH-), 3.59 (t, J = 5.8 Hz, 1H, -CH-).13C NMR (100 MHz, Acetone-d6) δ 166.12 (d, J = 254.9 Hz), 165.41, 161.48, 131.69 (d, J = 3.2 Hz), 131.32 (d, J = 10.1 Hz), 116.93 (d, J = 23.2 Hz), 80.99 (d, J = 169.1 Hz), 60.50, 32.06 (d, J = 22.0 Hz). 19F NMR (376 MHz, Acetone-d6) δ -104.33, -216.98. HRMS calculated for C11H11O4N2F2S2 [M + H]+ 337.01228, found 337.01169.
(5-((2-fluoroethyl)thio)-1,3,4-oxadiazol-2-yl)methyl 4-chlorobenzenesulfonate (4a-15). White liquid; yield, 80.5%; 1H NMR (400 MHz, CDCl3) δ 7.81-7.75 (m, 2H, Ar-H), 7.50-7.45 (m, 2H, Ar-H), 5.19 (s, 2H, -CH2-), 4.72 (t, J = 5.7 Hz, 1H, -CH-), 4.60 (t, J = 5.7 Hz, 1H, -CH-), 3.49 (t, J = 5.7 Hz, 1H, -CH-), 3.43 (t, J = 5.7 Hz, 1H, -CH-). 13C NMR (100 MHz, CDCl3) δ 166.17, 160.81, 141.37, 133.62, 129.81, 129.51, 80.62 (d, J = 172.1 Hz), 59.74, 32.33 (d, J = 22.2 Hz). 19F NMR (376 MHz, CDCl3) δ -215.71. HRMS calculated for C11H11O4N2ClFS2 [M + H]+ 352.98273, found 352.98209.
(5-((2-fluoroethyl)thio)-1,3,4-oxadiazol-2-yl)methyl 4-bromobenzenesulfonate (4a-16). White solid; m.p.: 90-91 °C; yield, 80.5%; 1H NMR (400 MHz, CDCl3) δ 7.73-7.68 (m, 2H, Ar-H), 7.68-7.61 (m, 2H, Ar-H), 5.19 (s, 2H, -CH2-), 4.72 (t, J = 5.7 Hz, 1H, -CH-), 4.61 (t, J = 5.7 Hz, 1H, -CH-), 3.49 (t, J = 5.7 Hz, 1H, -CH-), 3.44 (t, J = 5.7 Hz, 1H, -CH-); 13C NMR (100 MHz, CDCl3) δ 166.21, 160.80, 134.20, 132.83, 129.98, 129.55, 80.66 (d, J = 172.3 Hz), 59.78, 32.37 (d, J = 22.2 Hz). 19F NMR (376 MHz, CDCl3) δ -215.68. HRMS calculated for C11H11O4N2BrFS2 [M + H]+ 396.93222, found 396.93161.
(5-(methylthio)-1,3,4-oxadiazol-2-yl)methyl 4-fluorobenzoate (5a-2). White solid; m.p.: 53-54 °C; yield, 75.5%; 1H NMR (400 MHz, CDCl3) δ 8.02 (dd, J = 9.0, 5.4 Hz, 2H, Ar-H), 7.06 (t, J = 8.7 Hz, 2H, Ar-H), 5.42 (s, 2H, -CH2-), 2.67 (s, 3H, -CH3). 13C NMR (100 MHz, CDCl3) δ 166.99, 166.22 (d, J = 255.3 Hz), 164.52, 162.68, 132.66 (d, J = 9.5 Hz), 124.94 (d, J = 3.0 Hz), 115.85 (d, J = 22.1 Hz), 55.62, 14.61. 19F NMR (376 MHz, CDCl3) δ -104.05. HRMS calculated for C11H10O3N2FS [M + H]+ 269.03907, found 269.03897.
(5-(methylthio)-1,3,4-oxadiazol-2-yl)methyl 4-bromobenzoate (5a-4). White solid; m.p.: 82-83 °C; yield, 70.5%; 1H NMR (400 MHz, CDCl3) δ 7.86 (d, J = 8.7 Hz, 2H, Ar-H), 7.54 (d, J = 8.7 Hz, 2H, Ar-H), 5.42 (s, 2H, -CH2-), 2.68 (s, 3H, -CH3). 13C NMR (100 MHz, CDCl3) δ 167.01, 164.81, 162.57, 131.98, 131.48, 129.06, 127.59, 55.70, 14.61. HRMS calculated for C11H10O3N2BrS [M + H]+ 328.95900, found 328.95895.
(5-(methylthio)-1,3,4-oxadiazol-2-yl)methyl 4-methoxybenzoate (5a-5). White solid; m.p.: 35-36 °C; yield, 79.5%; 1H NMR (400 MHz, CDCl3) δ 8.04 (d, J = 9.0 Hz, 2H, Ar-H), 6.95 (d, J = 9.0 Hz, 2H, Ar-H), 5.49 (s, 2H, -CH2-), 3.89 (s, 3H, -CH3), 2.76 (s, 3H, -CH3).13C NMR (100 MHz, CDCl3) δ 166.83, 165.18, 163.97, 163.02, 132.13, 121.01, 113.84, 55.52, 55.28, 14.60. HRMS calculated for C12H13O4N2S [M + H]+ 281.05905, found 281.05884.
(5-(methylthio)-1,3,4-oxadiazol-2-yl)methyl dimethylcarbamate (5a-6). White liquid; yield, 81.5%; 1H NMR (400 MHz, CDCl3) δ 5.28 (s, 2H, -CH2-), 2.97 (s, 3H, -CH3), 2.95 (s, 3H, -CH3), 2.75 (s, 3H, -CH3). 13C NMR (100 MHz, CDCl3) δ 166.51, 163.49, 155.10, 56.13, 36.79, 36.05, 14.60. HRMS calculated for C7H12O3N3S [M + H]+ 218.05939, found 218.05922.
- Please replace "μmol/L" with more typical "μM".
Answer: Thanks and the suggestion has been taken care of. All "μmol/L" in 15 places was replaced with "μM" in our manuscript.
- Please remove the abbreviation "SEM" from keywords and explain it only once (the first time mentioned) in the text.
Answer: Thanks. Sure we had explain it in the Abstract first time mentioned and we should remove the abbreviation “SEM” from keywords.
Line 31, 125, 372 "(SEM)" was deleted.
- Please reduce number of significant figure in the reported results according to the experiment accuracy. For example, it is definitely incorrect to use 4 (and more!) significant figures for EC50 values or % in biological results. It is also unnecessary to report in the manuscript regression equation and r. They can find a better place in the Supplementary Data.
Answer: Thanks and according to your suggestion, we have reduced significant figure of EC50 to 3 numbers. And for your good suggestion we have put the following revised Table 2 into supplementary file. And a concise “Table 2” without regression equation and r were applied in the manuscript with an referring line “c The corresponding regression equation and r value for this EC50 were provided in supplementary data” in the bottom of the table illustration.
Table 2 in supplementary file:
Table 2. EC50 (µM) of some target compounds against Xanthomonas oryzae pv. oryzae and Xanthomonas axonopodis pv. citri a
|
Compd. |
Xanthomonas oryzae pv. oryzae |
Xanthomonas axonopodis pv. citri |
||||||||||
|
EC50 (µM) |
regression equation |
r |
EC50 (µM) |
regression equation |
r |
|||||||
|
4a-1 |
63.4±3.8 |
y = 2.70x + 0.13 |
0.988 |
114±7 |
y=1.85x + 1.18 |
0.977 |
||||||
|
4a-2 |
50.1±4.2 |
y = 3.21x - 0.45 |
0.988 |
95.8±4.6 |
y=2.10x + 1.04 |
0.984 |
||||||
|
4a-3 |
87.2±4.7 |
y = 2.87x - 0.58 |
0.964 |
132±7 |
y=1.56x + 1.69 |
0.976 |
||||||
|
4a-4 |
99.4±4.7 |
y = 3.11x - 1.22 |
0.966 |
155±6 |
y=1.32x + 2.10 |
0.979 |
||||||
|
4a-11 |
98.0±6.6 |
y = 3.22x - 1.41 |
0.953 |
/ |
/ |
/ |
||||||
|
4a-12 |
95.3±3.9 |
y = 3.15x - 1.23 |
0.954 |
/ |
/ |
/ |
||||||
|
4a-13 |
86.4±4.8 |
y = 3.22x - 1.23 |
0.951 |
/ |
/ |
/ |
||||||
|
4a-14 |
69.0±4.4 |
y = 3.52x - 1.47 |
0.976 |
/ |
/ |
/ |
||||||
|
4a-15 |
83.4±6.0 |
y = 3.32x - 1.37 |
0.990 |
/ |
/ |
/ |
||||||
|
4a-16 |
112±6 |
y = 2.60x - 0.33 |
0.946 |
/ |
/ |
/ |
||||||
|
Bismerthiazol b |
253±8 |
y = 2.03x + 0.12 |
0.969 |
274±9 |
y=1.76x + 0.70 |
0.978 |
||||||
|
Thiodiazole copper b |
467±15 |
y = 1.78x + 0.23 |
0.979 |
406±13 |
y=0.92x + 2.60 |
0.983 |
||||||
|
a The statistical analysis was conducted by ANOVA method at the condition of equal variances assumed (p > 0.05) and equal variances not assumed (p < 0.05); b The commercial agricultural antibacterial agents Bismerthiazol, and Thiodiazole copper were used as positive control. |
||||||||||||
Table 2 in manuscript
Table 2. EC50 (µM) of some target compounds against Xanthomonas oryzae pv. oryzae and Xanthomonas axonopodis pv. citri a
|
Compd. |
Xanthomonas oryzae pv. oryzae |
Xanthomonas axonopodis pv. citri |
|
EC50 (µM)c |
EC50 (µM)c |
|
|
4a-1 |
63.4±3.8 |
114.0±6.6 |
|
4a-2 |
50.1±4.2 |
95.8±4.6 |
|
4a-3 |
87.2±4.7 |
132.5±7.5 |
|
4a-4 |
99.4±4.7 |
155.2±5.8 |
|
4a-11 |
98.0±6.6 |
/ |
|
4a-12 |
95.3±3.9 |
/ |
|
4a-13 |
86.4±4.8 |
/ |
|
4a-14 |
69.0±4.4 |
/ |
|
4a-15 |
83.4±6.0 |
/ |
|
4a-16 |
112.5±6.0 |
/ |
|
Bismerthiazol b |
253.5±7.6 |
274.3±8.6 |
|
Thiodiazole copper b |
467.4±15.5 |
406.3±13.0 |
|
a The statistical analysis was conducted by ANOVA method at the condition of equal variances assumed (p > 0.05) and equal variances not assumed (p < 0.05); b The commercial agricultural antibacterial agents Bismerthiazol, and Thiodiazole copper were used as positive control. c The corresponding regression equation and r value for this EC50 were provided in supplementary data. |
||

Reviewer 3 Report
The manuscript can be published in present form.
Author Response
Reviewer 3:
The manuscript can be published in present form.
Answer: Thanks for your positive and helpful comments.
And in this round, there was still a few revisions in the manuscript as listed below for your reference.
- a) replacing "μmol/L" with more typical "μM",
- b) reducing significant figure of value % or EC50 to 3 numbers or with one digital after decimal point. And remove the original table 1 in supplementary file and using a concise table 1 without regression equation and r with an referring line “c The corresponding regression equation and r value for this EC50 were provided in supplementary data” in the bottom of the table illustration.
- c) modification in the structure characterization(for fluorinated compounds, coupling constant of J C F coupling indicated in their 13C NMR spectra were characterized).
